# Arbitrary-Order and Multichannel Optical Vortices with Simultaneous Amplitude and Phase Modulation on Plasmonic Metasurfaces

**DOI:** 10.3390/nano12193476

**Published:** 2022-10-04

**Authors:** Qing’an Sun, Wangying Yang, Lei Jin, Jingcheng Shangguan, Yilin Wang, Tong Cui, Kun Liang, Li Yu

**Affiliations:** 1State Key Laboratory of Information Photonics and Optical Communications, School of Science, Beijing University of Posts and Telecommunications, Beijing 100876, China; 2College of Mathematics and Physics, Beijing University of Chemical Technology, Beijing 100029, China; 3State Key Laboratory of Precision Measurement Technology and Instruments, Tsinghua University, Beijing 100084, China

**Keywords:** optical vortex, topological charge, spin–orbit interaction, geometric phase

## Abstract

The highly localized and uneven spatial distribution of the subwavelength light field in metal metasurfaces provides a promising means for the generation of optical vortices (OVs) with arbitrary topological charges. In this paper, a simple and reliable way for generating multichannel OVs on gold nanoporous metasurfaces is reported. The instantaneous field of arbitrary-order OVs can be regulated and concentrated on the same focal surface by adapting photonic spin–orbit interaction (SOI) and geometric phase. The focal ring energy distribution of OVs along the conical propagation path is accurately calculated, and the double phase of units induced by spin rotation is confirmed. Based on the parameter optimization of the nanohole arrangement, the simultaneous amplitude and phase modulation of multichannel OVs has been realized. Furthermore, the average multichannel signal-to-noise ratio exceeds 15 dB, which meets the requirements of high resolution and low crosstalk. Our study obtains broadband and efficient OVs, which can contribute to improving the capacity storage and security of optical information and possess great application prospects in beam shaping, optical tweezers, and communication coding.

## 1. Introduction

Optical vortices (OVs) in the visible band, with helical phase wavefront and orbital angular momentum (OAM) of multiple eigenstates, have broad application prospects in the fields of optical information multiplexing and optical manipulation [1,2,3]. Light with OAM has an $\exp\;\left(il\varphi \right)$ spiral phase described, where φ is the spatial azimuth, and l is the topological charge (TC) value of the OVs [4], which means that each photon can carry the OAM with the value of lℏ (ℏ is the reduced Plank constant). Compared with the spin angular momentum (SAM) with only two states, the OAM of photons implies more available modes of field states [5], which are orthogonal to each other [6]. Therefore, vortex beams have been widely used in the fields of high-capacity optical communication [7], nanophotonic detectors [8], optical manipulation of nanoparticles [9], near-field optical trapping [10], quantum memory [11], imaging [12], the phenomenon of bound states in the continuum [13], etc. In recent years, many methods have been proposed for the generation of OVs, such as using a spiral phase plate [14], nano resonant ring [15], plasmonic spiral shapes [16], metal nanorods, and grating nonstaggered metasurfaces [17]. However, due to a low signal-to-noise ratio (SNR) and low resolution [18], the OVs produced by those technologies cannot meet the requirements of integrated optical systems and miniaturized devices.

Optical metasurfaces are a kind of ultrathin planar optical element, which are composed of one or several layers of artificial structures assembled from traditional materials [19,20,21]. Through careful design of each element structure, the metal metasurfaces can provide an additional phase gradient or surface momentum [22,23]. Metal metasurfaces based on the Pancharatnam–Berry (PB) phase [24] or geometric phase can produce OVs through the spin–orbit interaction (SOI) [25], which are robust and nondispersive [26]. Although these surfaces have multiple functions, they still face some obvious challenges and shortcomings. On the one hand, discrete structural units interact with light on the metasurfaces, resulting in discontinuous wavefront modulation and relatively low efficiency [27]. On the other hand, the change in the spatial distribution of the metal structure increases the complexity of preparation, which makes the preparation of high-performance metasurfaces very time-consuming and expensive, especially in the visible spectrum range with short wavelengths.

In this study, we propose a single-layer metasurface to realize the generation of amplitude- and phase-dependent multichannel and arbitrary-order vortex beams. Due to the strong anisotropy and local field enhancement effect in nanostructures, there is a very high conversion effect between the SAM and the OAM of photons [28]. By using the finite-difference time-domain (FDTD) method, we numerically calculate the generation and propagation of the plasma vortex beams on metal metasurfaces. Different from the field localization effect of nanostructures, some specific optical localized states are bound in the near-field region and then partially scattered to the far field. Since the optical channels are composed of metal grids, near-field surface plasmon polaritons (SPPs), and far-field OVs [29], the total momentum of photons is conserved on the metasurfaces, and the angular momentum and energy of photons are conserved behind the metasurfaces. Based on these characteristics, adjusting the initial orientation angle of the nanohole groups can affect the phase of the transmitted light, and vortex beams working at broadband can be arbitrary-order multiplexed and demultiplexed [30]. Our plasmonic metasurface OV scheme may become a low-cost platform for large-scale optical manipulations, along with potential applications in optical detection, integrated components, imaging, and optical communications.

## 2. Design and Results

### 2.1. Diagram of Geometric Phase Nanohole Metasurfaces

The schematic of single-layer metal nanoholes is shown in Figure 1; the metal material is gold, whose experimental data is referred to in [31]. In the FDTD simulation program, the dielectric constant of Au also draws on the Drude model, while the refractive index of the glass substrate is fixed at 1.5 [32]. Figure 1a shows the spin–orbit interaction of the left-hand circularly polarized (LCP) light passing through the metasurface to produce OVs. The two illustrations above the model are the focal ring energy and phase distributions of the output fields, respectively, with the position of the receiving surface at $z_{1}$ = 2000 nm. Figure 1b is a structural diagram of a single metal nanohole. The dimensions of each unit cell in plane are 300 × 300 nm^2^, and the thickness of the metal film and the glass substrate is 80 nm and 2 μm, respectively. All the geometric parameters are listed in Table 1. The long half-axis length $a_{1}$ of the nanohole is 95 nm and the short half-axis length $b_{1}$ is 50 nm. Generally, simple and compact structural units, whose phase response is independent of the wavelength of incident light, can realize the function of “colorless dispersion” [33,34]. Notably, Figure 1c shows the transmission spectra affected by the change in the short axis radius of the nanoholes, with a peak at 530 nm on the left. However, in order to pursue a higher conversion efficiency of the SOI, the working band is at a wavelength with a larger conversion response, with a peak at 660 nm on the right. Moreover, it is also required that the structural units have different electric field responses to LCP and RCP light in the working band and induce additional phases from the perspective of nanopore orientation, as shown in Figure 1d,e. Therefore, the central working wavelength of the metasurfaces is around 632 nm, and the bandwidth is approximately 150 nm, to meet the requirement that the transformation efficiency of the SOI is greater than 50%.

As shown in Figure 2, circularly polarized light perpendicularly incident on the gold etching film from below generates an asymmetric surface plasmon field. The orientation angle *θ* of each metal elliptical nanohole is the rotation angle of its major axis relative to the *y* axis, which can take a value from 0 to 2π. These nanoholes with a total number of m are used to form a ring with radius R. When passing through the metasurface, part of the SAM can be transformed into the OAM through the SOI of photons. After conversion, the radius of the focal ring of the optical vortex increases with the propagation distance, and at the same time the phase plane spirals forward, as shown in Figure 2b,c. Moreover, in Figure 2d, by controlling the rotation angles of elliptical nanoholes relative to the Y direction, a complete phase transition of −2*π*~2*π* can be obtained in a wide band. More importantly, this three-dimensional mapping measurement can clearly show the dynamic evolution process of the wavefront, which is helpful for intuitively understanding the change process of the photon spin–orbit interaction from the near field to the far field in the process of vortex light generation.

Rotating the nanostructure by an angle *θ* relative to the independent frame will introduce a geometric phase. Its phase change will produce the same or opposite circular polarization as the incident light, which is shown by the fact that the induced phase is twice the orientation angle *θ*. A certain number of metal nanoholes are arranged in an array on the metasurfaces, and the left/right circularly polarized light incident through the metasurfaces can produce optical vortices of different orders. The construction equation for determining the array arrangement mode according to the topological charge of the required optical vortex is as follows:(1)2πl+Φ=2m×Δθ ,
where θ is the rotation angle of the metal elliptical nanohole relative to the *y* axis, and Δθ is the difference in the rotation angles between adjacent nanoholes with circular distribution. Φ is added to offset the loss angle of SPPs excited by different nanometal structures on the helical phase distribution of scattered light, and $\Phi /k_{SPPs}=2\pi R-mP-m\left(a_{1}+b_{1}\right)/2$, where P is the simplified period of the nanoholes. m is the number of pores around the center of the nanostructure and can be taken as 8, 12, 18, 24.

The nanostructures with a ring distribution can reflect the physical essence that the additional phase comes from the transformation of a coordinate system better than those with an array distribution [35]. Although it is more difficult in experiments to adjust the orientation angle of the nanoholes than a single plasma helix [9], the higher conversion efficiency of the SOI and the morphology of the higher-order OVs accorded with our expectations.

Figure 3 is a schematic diagram of the realization of OVs on the back focal plane of the metasurface. This demonstration method of independent first- to eighth-order OVs is more intuitive, and each beam with the phase of $exp\left(il\varphi \right)$ shows macro and typical characteristics: the phase front forms a spiral dislocation or vortex on the beam axis. As shown in Figure 3ai–hi, the focal ring energy is actually the power of the cross-polarized output field. This arithmetic, $P_{cr}\approx {\left(E_{x}^{out}+i\sigma E_{y}^{out}\right)}^{2}/2$ can be written directly in our simulation program [36]. The transmitted electric field data it needs are shown in Figure 3aii–hii. The original numerical solver of commercial FDTD solutions is used (see details in Appendix A). The simulation results show that the first- to eighth-order OVs can be realized through similar structures, which is simple and reliable. As shown in Figure 3eiii–hiii, on the larger propagation cross section, or at the center of the high-order OVs, the optical vortex transmutation may also be observed [37].

Subsurface micro/nano machining technology provides a new development platform [37] for geometric metasurfaces. The adjustment of the geometric phase to the scattered light only depends on the local rotation of the nanostructures, which is wavelength-independent. Therefore, our schemes have good tolerance in sample preparation.

### 2.2. Theoretical Explanation

For the spin–orbit conversion effect, the internal orbital angular momentum carried by the far-field vortex beam is generated by the phase gradient distribution of the light field in the near field. The SOI of higher-order photons in rotationally symmetric structures was studied. Referring to the transverse distribution expression of the complex general solutions of the Laguerre–Gaussian (LG) beam mode [38], we assume that that the waist radius $w_{0}$ of the emitted light is known, and the electric field can be written in the cylindrical coordinate system:(2)$$E\left(r,\varphi ,z\right)=\left\{\begin{array}{c} \frac{w_{0}}{w_{l}}{\left(\sqrt{2}\frac{r}{w_{l}}\right)}^{\left|l\right|}\;(e^{i\left(l\varphi -kz\right)-\frac{r^{2}}{w_{l}^{2}}})\;z\geq \delta_{d}\\ 2\frac{r^{2}}{w_{0}^{2}}(e^{i\left(l\varphi -k_{SPP}z\right)-\frac{r^{2}}{w_{0}^{2}}})\;\;\;0\leq z<\delta_{d} \end{array}\right.,$$
where $\delta_{d}$ is the skin depth of the near-field plasma vortex at the metal–dielectric interface; k=2π/λ is the wave vector; and $w_{l}\left(z\right)=w_{0}\sqrt{l^{2}+{\left(\frac{z}{f_{0}}\right)}^{2}}$ is the waist radius of higher-order OVs. $f_{0}$ is not only the focal length but also related to the Rayleigh range [39], in which the wavefront of the OVs can be regarded as equivalent to a plane. It is worth noting that for a vortex beam there must be one or more phase singularities, and the phase of the light field at these positions cannot be defined. At this time, to make the whole light field meaningful, the amplitude of the electric field at the phase singularities must be zero, otherwise it cannot exist physically. At this time, the phase distribution after being applied to the metasurfaces is:(3)$$\varphi_{l}\left(r,\varphi ,z\right)=k\sqrt{r^{2}+z^{2}}+l\varphi +k_{SPP}\cdot R+2\sigma \theta_{1}.$$

Equation (3) implies the propagation path of near-field plasma vortices and far-field optical vortices. σ is determined by the circular polarization state of the incident light; and $\theta_{1}$ is the initial orientation angle of the nanohole numbered 1 relative to the *y* axis, whose optimal value makes it possible to regulate the receiving phase of multiple groups of nanohole arrays.

Corresponding to the simulation results in Figure 3ai–hi, the focal rings of optical vortices of different orders propagate in the free space behind the metasurfaces, satisfying the conservation of energy and angular momentum. However, its shape is not like the standard circle shown in previous studies [22,40] but an irregular polygon, which may be caused by our oversimplification of the structure. As shown in Figure 4a–h, the simulation phenomenon is verified by mathematical calculation in the cross-sectional electric field distribution diagram with the energy or wave vector as the coordinate axis. The theoretical calculation shows the electric potential line projected at the cross section of the arbitrary-order OVs traveling within two wavelengths, which exactly conforms to the simulation results in Figure 3i–ii. In order that the instantaneous field of arbitrary-order OVs can be adjusted and concentrated on the same focal surface, the propagation paths of OVs of different orders calculated according to our theory are shown in Figure 4i–l. Because the simulation results in Figure 3 are all located at $z_{1}$ = 2000 nm, the theoretical arithmetic also calculates up to this distance.

## 3. Further Exploration

### 3.1. Arbitrary-Order and Multichannel Methods to Solve the Problems of Low Resolution and Large Crosstalk

The metasurface of the metal hole array has good adaptability to the polarization direction of incident light. When multiple polarization directions or circularly polarized light are incident, the outgoing light can maintain good vortex characteristics and has good adjustability to the polarization direction of incident light. The above structure generating multiorder OVs is integrated on the same SiO_2_ substrate (2 μm thick) shown in Figure 5. After the circularly polarized light passes through, four noninterference OVs are formed, and their light field distribution and phase distribution are shown in Figure 5b–e.

The numerical and simulation results are shown in Figure 5, in which all positions of the receiving section are at $z_{1}$ = 2000 nm. When the total phase difference between the plasma mode and the photon mode is greater than π/2, as shown in Figure 5a, the four-channel phase transition can clearly show the cycle along the disk, and a uniform phase distribution is found in the scattering cross section. In this region, the characteristics of the propagating OVs with topological charge numbers of (1,2,3,4) can be more clearly observed in Figure 5b,c. What is more surprising is that in the example in Figure 5d,e we obtained multichannel optical vortices of the same order with different optical field energies and on-demand phase differences by adjusting the initial orientation angle of the nanoholes that generate the same-order OVs, with topological charge numbers of (2,3,2,3).

In particular, it can be seen from Figure 5a that ${\left|E_{l=8}\right|}^{2}>$
${\left|E_{l=6}\right|}^{2}>$
${\left|E_{l=2}\right|}^{2}>{\left|E_{l=4}\right|}^{2}$. Originally compared with Equation (2) and the simulation results of Figure 3, the focal ring radius of the low- to high-order optical vortex gradually increases and the energy gradually decreases, which is consistent with Figure 5b,c. However, through our design of the optimal initial orientation angle of the nanopore group, the power of the eighth-order optical vortex is the largest under the same radius R. This is because the actual value range of the initial orientation angle of the elliptical nanoholes is $\left|\theta_{1}\right|\leq \frac{l\pi }{2m}$ or $\left|\theta_{1}-\frac{n\pi }{2}\right|\leq \frac{l\pi }{2m}$. That is, the range of the angle $\theta_{1}$ of the nanoholes constituting the higher-order OVs is larger, and it is easier to adjust the amplitude of the light emitted from the channel. In contrast, if the orientation angle of the nanoholes that form the low-order OVs always induces scattering from the center to the outside, the received light energy will also be greatly reduced. Referring to the calculation of the phase in Equation (3), the initial orientation angles of different nanopore groups can also be introduced. As shown in Figure 5d,e, the optical field amplitudes of two second-order OV channels are quite different; the central phase difference of the two third-order OV channels is π/3.

Because this kind of structure is very simple and compact, in accordance with the results in Figure 5, the average multichannel signal-to-noise ratio $SNR=10\times \mathrm{lg}\left(P_{cr}/P_{co}\right)$ exceeds 15 dB, where $P_{co}\approx \frac{1}{2}{\left(E_{x}^{out}-i\sigma E_{y}^{out}\right)}^{2}$ is the copolarized output field (with the same spin as the incidence). This means that a large amount of transmitted energy is carried by the cross-polarized OV component, and most of the copolarized incident light is shielded by the opaque gold film, which makes our background noise lower. In addition, the resulting OV size is also small, with an average diameter of only 0.95 μm, and the center distance of each focal ring is 2.2 μm, which is also much smaller than previous articles of the same type [14,41]. Therefore, this paper obtains a smaller OV beam and a larger polarization conversion rate of approximately 80% (taking the transmitted light field energy as the denominator parameter). The annular distribution of nanoholes can act as a near-field phase modulator, providing a simple, reliable, and efficient method for the production of 3D vortex beams. Numerical simulation shows that this phase and amplitude manipulation can be used to create a well-defined vortex beam array and realize the offset of an arbitrary output signal through the same source of off-axis technology, which has a broad application prospect in multiplexing.

### 3.2. Design of Two-Channel Vortex Generator with the Same Propagation Direction

For the purpose of exploring the light–matter interaction on the homologous multichannel metasurface, the near field and far field are set in identical propagation directions and different focal points. With the same structure, the larger the radius of the surrounding nanoholes, the farther the focus of the OVs; and the higher the order of the OVs, the closer the focus, thus separating each channel. As shown in Figure 6, the outer ring (with a large radius) is a low-order OV, and the inner ring (with a small radius) is a high-order OV. The light field energies of near-field sixth-order OVs and far-field second-order OVs are shown in Figure 6b,e, respectively.

The metal elliptical nanoholes of the manufactured concentric disk are shown in Figure 6a. The inner and outer radii $R_{1},\;R_{2}$ were chosen to be 1.26 and 2.34 μm. The electric field of the scattered light on the X-Y plane through the metasurface is shown in Figure 6d. The phase distributions at different distances ($z_{2}$ = 1500 nm and $z_{3}$ = 5000 nm) above the substrate were measured, as shown in Figure 6c,f. The separated phase distributions were 12π and 4π, showing OV beams with topological charge numbers of $l_{near}=6,\;l_{far}=2$. The focal ring radius of high-order OVs increases along the Z direction; on the contrary, the focal ring radius of low-order OVs decreases along the Z direction, which facilitates the reception of two channels and low crosstalk. Meanwhile, the beam size and divergence angle of the bi-channel vortex beams can be calculated directly. Because the vortex beam with orbital angular momentum has spatial orthogonality, the designed wavelength and polarization multiplexing metasurfaces expand the research scope of subsurface compact multifunctional photonics and may have a wide range of applications in information security and anticounterfeiting [42]

## 4. Conclusions

In our study, optical vortices were generated by the spin–orbit interaction of LCP light through the metasurface of single-layer metal nanoholes. Starting from the general solution of the LG beams, we numerically solved the electric field expression of OVs and the polygonal energy diagram of focal rings. By controlling the nanohole arrangement based on the theoretical equations, arbitrary manipulations of OAM superposition in four channels were observed, and the optimized metasurfaces enabled high-quality OVs. The near-field amplitude and phase distribution of the OV array generated by circumferential nanoholes were measured, with an average multichannel resolution of more than 15 dB, which provides an effective way to increase information capacity and safety. Due to the simplicity and completeness of our design, this study may be of great significance to the theoretical research of geometric phases and the practical application of OAM equipment.

## Figures and Tables

**Figure 1 nanomaterials-12-03476-f001:**
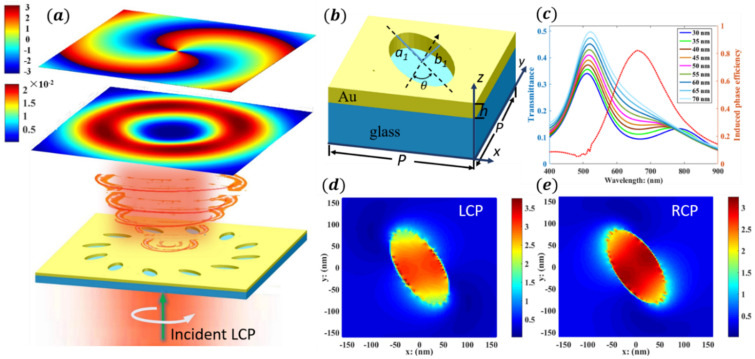
(**a**) The LCP light passes through the metasurface of single-layer metal nanoholes and generates OVs based on SOI. (**b**) Schematic of single elliptical nanohole structure. (**c**) Left: transmission spectra affected by the change in the short axis radius of single elliptical nanohole. Right: the efficiency of transmitted light carrying additional phase at different wavelengths. (**d**,**e**) The resonance electric field |E| generated by the incidence of 632 nm LCP and RCP light, respectively.

**Figure 2 nanomaterials-12-03476-f002:**
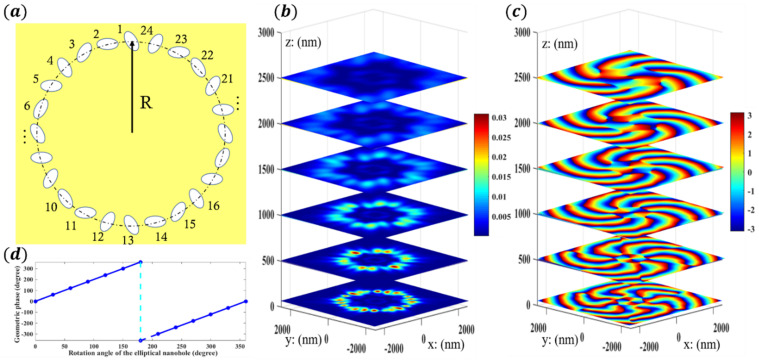
Design schematic of the geometric metasurface and a display of the eighth-order optical vortices’ (OVs) morphology. (**a**) The arrangement scheme of metal elliptical nanoholes inducing high topological charges. (**b**,**c**) The distribution of the focal ring energy and the spatial phase change on the propagation path of light scattered behind the metasurface of single-layer metal nanoholes, respectively. (**d**) The geometric phases produced by different rotation angles of elliptical nanoholes.

**Figure 3 nanomaterials-12-03476-f003:**
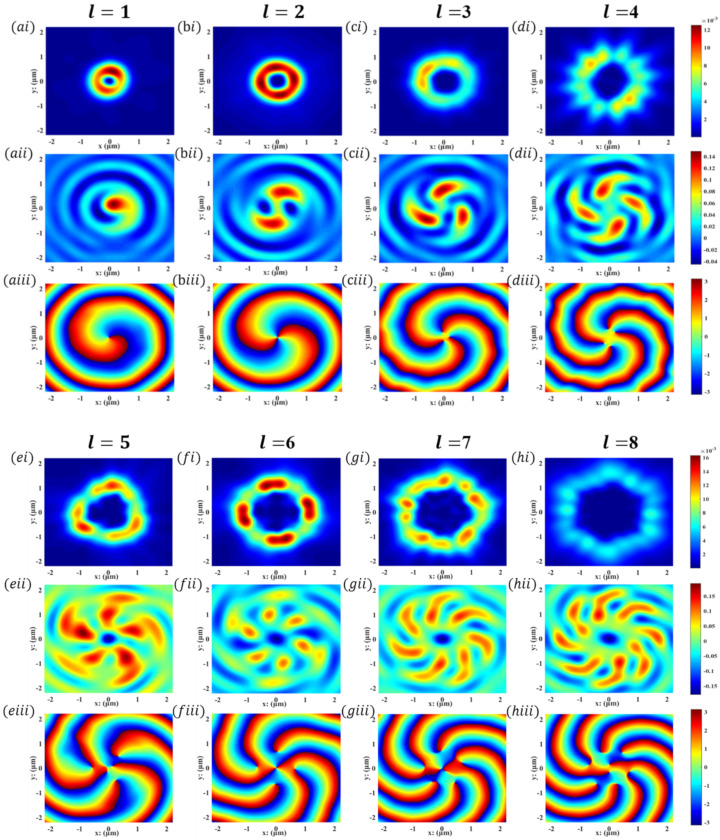
(**a**–**h**) Generation of OVs with independent first- to eighth-order. (**i**) The focal ring energy distribution of transmitted light, or actually the power of the cross-polarized output field. (**ii**) The distribution of the transmitted electric field $E_{x}$ in the x direction. (**iii**) The phase distribution of the transmitted light. In all simulations in this figure, the incident light is set at 632 nm, and all data are collected at $z_{1}$ = 2000 nm.

**Figure 4 nanomaterials-12-03476-f004:**
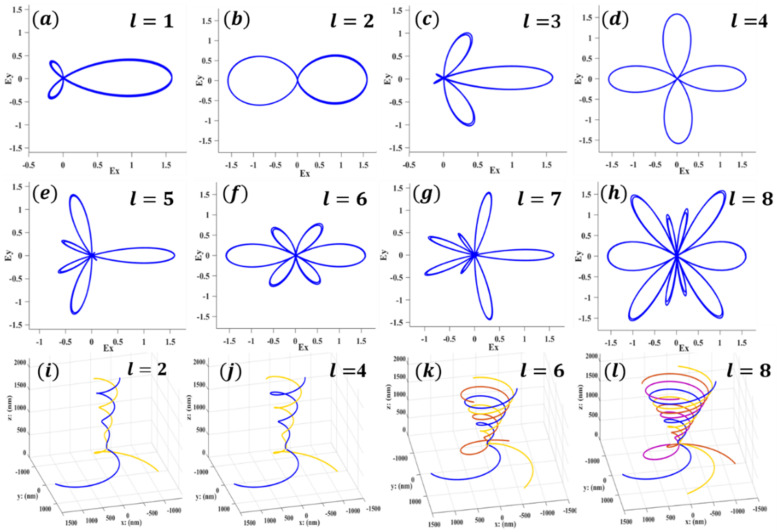
(**a**–**h**) Electric field distribution on the propagation cross section of optical vortex focal rings with different topological charges calculated according to our theory. (**i**–**l**) The propagation paths of optical vortices of different orders calculated according to the theory, and z=0 is the upper surface of metal.

**Figure 5 nanomaterials-12-03476-f005:**
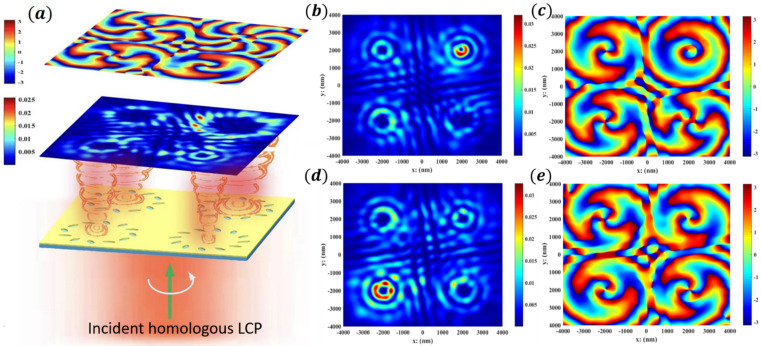
(**a**) Schematic diagram of homologous four-channel multi-order (8,6,4,2) OV array. (**b**,**c**) Scattering energy diagram and propagation section phase distribution diagram of homologous four-channel multi-order (1,2,3,4) OVs. (**d**,**e**) Scattering energy diagram and propagation section phase distribution diagram of homologous four-channel multi-order (2,3,2,3) OVs.

**Figure 6 nanomaterials-12-03476-f006:**
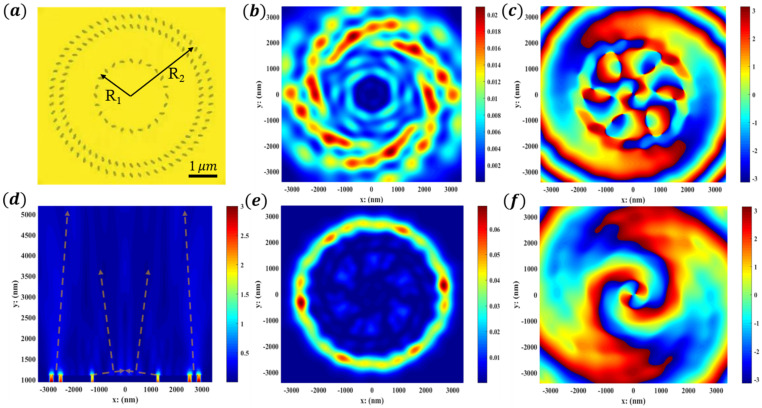
Realization of homologous bi-channel OVs in the near field and far field. (**a**) The array distribution of metal elliptical nanoholes. (**b**,**c**) Focal ring energy and phase distributions of near-field high-order (l=6) OV at $z_{2}$ = 1500 nm. (**d**) The electric field distribution of scattered light on the X-Y plane, with the metasurface at the bottom. The dotted lines show the propagation path of OVs. (**e**,**f**) Focal ring energy and phase distributions of far-field low-order (l=2) OV at $z_{3}$ = 5000 nm.

**Table 1 nanomaterials-12-03476-t001:** Geometric parameters of the metasurfaces.

P [nm]	$a_{1}$ [nm]	$b_{1}$ [nm]	h [nm]	H (Glass) [μm]	R (When m = 12) [μm]	R (When m = 24) [μm]
300	95	50	80	2	0.68	1.26

## Data Availability

The data that support the findings of this study are available from the corresponding author upon reasonable request.

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
