# Peer review of "Arbitrary-Order and Multichannel Optical Vortices with Simultaneous Amplitude and Phase Modulation on Plasmonic Metasurfaces"

_nanomaterials, 2022, doi:10.3390/nano12193476_

Round 1

Reviewer 1 Report

In this work, using FDTD simulation, the authors showed that a set of elliptical nanoholes in a gold film located on a circle of a certain radius will form an optical vortex at a distance of 2 μm from the surface. The fact that a set of various holes (nanoantennas) in a gold film, when illuminated with light with circular polarization, generates an optical vortex has long been known [25, 30]. What is new in this work is that the holes are elliptical in shape. The work can be published after the authors take into account the comments.

Comments

1. In Fig. 2d, there is a graph of the dependence of the phase on the number of nanoholes. The authors should add a graph of the dependence of the phase on the angle of rotation of the major axis of the nanoholes ellipse.

2. Equation (1) should be commented in detail, since the value of Δθ is undefined.

3. In the equation in line 131, the phase Ф does not depend on the angle φ, and in equation (1), the phase Ф depends on the angle φ. This contradiction must be eliminated.

4. Line 155 says that equation (2) describes the Laguerre-Gauss (LG) modes. For the LG mode, the degree of the radial variable r^l must be equal to the topological charge of the phase lφ. And in equation (2) r^2 and r^(l+2), so equation (2) does not describe the LG mode. In addition, in (2) for z>δ, the Gaussian exponent is omitted.

5. In line 159, the waist radius w is written incorrectly, since it turns out that it is equal to the dimensionless topological charge l.

6. In equation (3), the first term of the quadratic phase is written incorrectly, since the radius of curvature of the front depends on z in a different way. It turns out that already on the surface at z=0 the wave front has a parabolic curvature.

7. On line 166-167 it is written that the initial angle θ allows you to adjust the phase. But the constant addition 2σθ to phase (3) has no physical meaning, since the phase is defined up to an arbitrary constant.

8. Figure 4 if it is derived from Equation (2) is incorrect. Therefore, Fig. 4 should either be deleted or its meaning should be explained in detail. What do these figures mean? Why, for even l, do the figures have 2 times more petals than l? And why do the paintings at l=6 and l=8 have an equal number of petals?

Author Response

Thanks a lot for your kindly reviewing and professional comments, which are valuable and helpful for improving our paper. We have read the comments carefully and have made corrections according to your comments and requirements. 

Reviewer 2 Report

The paper presents a plasmonic structure consisting of non-hole arrays to produce optical vortices. The work is simulation based and presents good numerical results. The simulation results show generation of different orders of optical vortices. The work is interesting and should be considered for publication. However, the presentation quality of the manuscript needs to be significantly improved. A lot of key information are either missing or discussed inadequately. Many of the important design and output parameter values should have listed in Tables instead of being scattered in the text. Very little information about the simulation tool and simulation setup parameters are included. The motivation of the work is not justified sufficiently. For example, it is stated in line 45 that current technologies cannot meet the requirements. However, no explanation is provided nor any references cited. The design requirements are not stated. How the geometric parameters were selected has not been discussed (like the dimensions of the nano-hole). The reference section is inadequate as well. I would recommend the authors to present their work in a more organized way that makes the motivation, the design criterion, the design process, and the significance of the results more clearly. After this modifications are made, the paper should be suitable for publication. Some of my specific comments are listed below:

  1. What was the thickness of the metal film used for the simulation (h in Fig. 1b)? I could not locate the value. It might be good to list all the geometric parameters in a table.

  2. Please discuss which model was used for the refractive index of gold. Were the values taken from Johnson and Christy other tabulated values? Was closed form expressions like the Drude model used? As the plasmonic response depends on the refractive index, it is necessary to include the details so that the results can be reproduced. Along with gold, refractive indices of all other the relevant materials should be mentioned in the paper.

  3. What numerical solver was used to produce the results? Was it a custom FDTD code? Or a commercial FDTD solver? Please include details.

  4. The proposed design involves having circular arrays of nano-structures each rotated by some angle. Line 216 and 217 shows a limit on the angle. However, the exact angle used in the simulation was not clearly defined. Please mention this.

  5. Although the work is purely numerical, some practical limitations of the design should be mentioned. For example, fabricating the structures with precise angular orientation can be difficult. Please briefly discuss this.

  6. It is possible to generate orbital angular momentum (OAM) using single plasmonic structures instead of using arrays of smaller nano-structures. For example, plasmonic spiral shapes have been successfully used to generate OAM. Tsai et. al have reported an excellent work (https://doi.org/10.1021/nl403608a) regarding this phenomenon. The force profile of such structures have been discussed in detail in https://doi.org/10.1103/PhysRevA.100.013857. It would be good to mention this works in the introduction section. Also, please briefly discuss the relative advantages/disadvantages of using an array instead of a single structure.

  7. One of the main applications of plasmonic focusing of light is optical manipulation of micro and nano-particles. This should be mentioned when listing applications in line 40-41. Along with the afore mentioned Tsai et. al paper, a few of Prof. K Crozier’s and Prof. Lambertus Hesselink’s papers on near-field optical trapping in a non-conservative force should be cited.

  8. In the introduction section (line 70-73), potential multiplexing capabilities are hinted. However, this has not been demonstrated. This was not elaborated later either. Please comment.

  9. The y axis of Fig. 1C has been clipped. Please include the full the y axis range so that the curves can be read.

  10. It is unclear what the different curves in Fig. 1C represents. The legend details were not mentioned in the text nor in the figure caption. Please elaborate.

  11. Line 88-91 states that wavelength was shifted higher (from the transmission peak) in order to the have a larger resonance response. Does this mean that the resonance response peak and the transmission peak occur at different wavelengths? This would be unusual. If this is the case, then please produce the wavelength vs plasmon response amplitude plot.

  12. In section 2.1, please define bandwidth clearly. Is it referring to the wavelength range of the excitation light? Or is it referring to the wavelength range at which the structure produces large transmission (like a Half-Power Bandwidth). Such details need to be made clear.

  13. The axis text in Fig. 3 are too small to be legible. Also, please mention at what wavelength the simulation was performed.

Author Response

(The authors gave the same response as above.)

Round 2

Reviewer 2 Report

The paper has been improved.